# Molecular and Functional Imaging Studies of Psychedelic Drug Action in Animals and Humans

**DOI:** 10.3390/molecules26092451

**Published:** 2021-04-22

**Authors:** Paul Cumming, Milan Scheidegger, Dario Dornbierer, Mikael Palner, Boris B. Quednow, Chantal Martin-Soelch

**Affiliations:** 1Department of Nuclear Medicine, Bern University Hospital, CH-3010 Bern, Switzerland; 2School of Psychology and Counselling, Queensland University of Technology, Brisbane 4059, Australia; 3Department of Psychiatry, Psychotherapy and Psychosomatics, Psychiatric Hospital of the University of Zurich, CH-8032 Zurich, Switzerland; milan.scheidegger@bli.uzh.ch (M.S.); dornbierer@pharma.uzh.ch (D.D.); boris.quednow@bli.uzh.ch (B.B.Q.); 4Odense Department of Clinical Research, University of Southern Denmark, DK-5000 Odense, Denmark; mikael.palner@gmail.com; 5Department of Nuclear Medicine, Odense University Hospital, DK-5000 Odense, Denmark; 6Neurobiology Research Unit, Copenhagen University Hospital, DK-2100 Copenhagen, Denmark; 7Neuroscience Center Zurich, University of Zurich and Swiss Federal Institute of Technology Zurich, CH-8058 Zurich, Switzerland; 8Department of Psychology, University of Fribourg, CH-1700 Fribourg, Switzerland; chantal.martinsoelch@unifr.ch

**Keywords:** hallucinogens, molecular imaging, PET, SPECT, serotonin receptors

## Abstract

Hallucinogens are a loosely defined group of compounds including LSD, *N,N*-dimethyltryptamines, mescaline, psilocybin/psilocin, and 2,5-dimethoxy-4-methamphetamine (DOM), which can evoke intense visual and emotional experiences. We are witnessing a renaissance of research interest in hallucinogens, driven by increasing awareness of their psychotherapeutic potential. As such, we now present a narrative review of the literature on hallucinogen binding in vitro and ex vivo, and the various molecular imaging studies with positron emission tomography (PET) or single photon emission computer tomography (SPECT). In general, molecular imaging can depict the uptake and binding distribution of labelled hallucinogenic compounds or their congeners in the brain, as was shown in an early PET study with *N*^1^-([^11^C]-methyl)-2-bromo-LSD ([^11^C]-MBL); displacement with the non-radioactive competitor ketanserin confirmed that the majority of [^11^C]-MBL specific binding was to serotonin 5-HT_2A_ receptors. However, interactions at serotonin 5HT_1A_ and other classes of receptors and pleotropic effects on second messenger pathways may contribute to the particular experiential phenomenologies of LSD and other hallucinogenic compounds. Other salient aspects of hallucinogen action include permeability to the blood–brain barrier, the rates of metabolism and elimination, and the formation of active metabolites. Despite the maturation of radiochemistry and molecular imaging in recent years, there has been only a handful of PET or SPECT studies of radiolabeled hallucinogens, most recently using the 5-HT_2A/2C_ agonist *N*-(2[^11^CH_3_O]-methoxybenzyl)-2,5-dimethoxy- 4-bromophenethylamine ([^11^C]Cimbi-36). In addition to PET studies of target engagement at neuroreceptors and transporters, there is a small number of studies on the effects of hallucinogenic compounds on cerebral perfusion ([^15^O]-water) or metabolism ([^18^F]-fluorodeoxyglucose/FDG). There remains considerable scope for basic imaging research on the sites of interaction of hallucinogens and their cerebrometabolic effects; we expect that hybrid imaging with PET in conjunction with functional magnetic resonance imaging (fMRI) should provide especially useful for the next phase of this research.

Contents:
IntroductionBinding Sites of Hallucinogens in Vitro
2.1.The Nature of Agonist-Receptor Interactions2.2.Affinities of LSD at Neuroreceptors in Vitro2.3.Affinities of Hallucinogenic Phenylethylamines in Vitro2.4.Affinities of Hallucinogenic Tryptamines in Vitro2.5.The Strange Case of IbogaineEx vivo/In vitro Binding Studies with Hallucinogens
3.1.LSD Derivatives3.2.Phenyethylamine DerivativesMolecular Imaging Studies in Vivo with Hallucinogens
4.1.LSD Derivatives4.2.Phenylethylamine Derivatives4.3.Tryptamine Derivatives4.4.Competition from Hallucinogens at Dopamine Receptors in Vivo4.5.Competition from Hallucinogens at Serotonin Receptors in VivoMetabolism of Hallucinogenic Compounds and Tracers
4.1.LSD5.2.Phenylethylamine Derivatives5.3.Tryptamine DerivativesAyahuasca and PharmahuascaEffects of Hallucinogens on Energy Metabolism and Perfusion
7.1.Cerebral Glucose Metabolic Rate7.2.Cerebral Blood FlowGeneral Conclusions

References

## 1. Introduction

Structurally diverse ergolines, phenylethylamines, and tryptamines known collectively as hallucinogens induce perceptual and affective changes, extending from sensory distortions (illusions) to sensing of non-existent objects (hallucinations), with varying degree of control over or insight into the altered state. The sites of hallucinogen binding and action in the central nervous system are amenable to study by molecular imaging with positron emission tomography (PET) or single photon emission computer tomography (SPECT), and can now be studied by functional magnetic resonance imaging (fMRI) of cerebral perfusion and connectivity [1]. A prohibition against hallucinogen research established in many countries the 1970s was until recently an impediment to progress in our understanding of the phenomenology and physiology of hallucinogen action [2,3]; PubMed hits for the search term “hallucinogen” peaked in 1974, troughed around 1990, and have sustained a high level since 2010. Analysis of the literature in the past decade shows a shift in emphasis from preclinical studies especially of lysergic acid diethylamide (LSD) (**1**) towards more clinical applications, especially involving psilocybin (**2**) [4]. Indeed, there is now considerable interest in exploring the psychotherapeutic potential of hallucinogens, with 13 trials of psilocybin (**2**) undertaken in 2020 alone [5]. The renewed exploration of hallucinogens in a therapeutic setting raises important ethical and scientific consideration, and highlights the need for basic research on the action of hallucinogens in human brain [6,7]. The only recent review on imaging of hallucinogen actions focusses mainly on the fMRI literature [8]. Given this background, we now present a narrative review of the present state of the molecular imaging literature on hallucinogenic molecules. We emphasize first the selectivity and affinity of hallucinogenic compounds for serotonin receptors in vitro, discuss their fitness as radioligands for autoradiographic binding studies, and finally review the rather sparse literature on PET and SPECT studies with hallucinogens. Our aim is to extract general principles from the available results, and identify topics for future research in this domain.

While various ancient peoples knew about plant- and animal-derived hallucinogens, the modern era of interest in hallucinogens began with the accidental discovery of the mind-altering effects of the ergot derivative LSD (**1**) (Figure 1). The extraordinary nature of experiences provoked by hallucinogens naturally motivated scientists to seek an understanding of their psychopharmacology and mechanisms of action. Hallucinogens belong to a variety of structural classes, including ergolines, phenylethylamines, and tryptamines (Figure 1, Figure 2, Figure 3 and Figure 5). As presented below, there is a general agreement that hallucinogens of the psychedelics class are agonists at serotonin 5HT_2A_ receptors; although drugs of diverse other pharmacological classes can also induce hallucinations, we mainly confine this review to serotoninergic substances. However, the full spectrum of a given hallucinogen’s action may well entail actions at other serotonin receptors as well as receptors of dopamine, noradrenaline, histamine, trace amines, and neurotransmitter uptake sites (e.g., [9,10]). The spectrum of target engagement must somehow account for the overlapping and distinct aspects of the phenomenologies of different hallucinogenic drugs. Indeed, small structural modifications of certain molecules can decisively influence their hallucinogenic potency. For example, unlike *N*,*N*-diethyltryptamine (*N*,*N*-DET, **3**), the corresponding diethyl compound, 6-fluoro-*N*,*N-*diethyltryptamine (**4**) is without hallucinogenic action in humans [11]. Similarly, 2-bromo-LSD (BOL-148, **5**), despite its engagement with serotonin 5HT_2A_ receptors, does not evoke hallucinations [12], except perhaps in rare cases [13]. Recent investigation of this phenomenon indicated that halogenation of *N*,*N*-DET (**3**) did not alter its affinity for serotonin 5HT_2A_ receptors or various other receptor types, but disabled the intracellular response to agonism, i.e., stimulation of phosphotidyl inositol (PI) turnover [14], which may be a necessary property of effective hallucinogens. However, hallucinogenic activity is not a simple binary phenomenon, but encompasses a range of visual and sensory experiences. For example, mescaline (**6**) tends to evoke a characteristic visual experience of “geometricization” of three-dimensional objects, as is depicted in certain Amerindian art traditions. Visual hallucinations hint at a pharmacological action in the visual cortex [15], which is a theme amenable to analysis using a neural network approach [16]. It remains unclear how the particulars of mescaline (**6**) pharmacology might account for its greater propensity to evoke a specific type of visual experience. Despite the broadly overlapping serotonin receptor binding profiles of hallucinogenic tryptamines, interactions at other receptor types may contribute to their overall psychopharmacology or the particular phenomenology of hallucinogenic experiences [17]. Indeed, pretreatment of volunteers with the selective 5HT_2A/C_ antagonist ketanserin (**7**), while largely abolishing the hallucinations elicited by psilocybin (**2**), did not attenuate binocular rivalry switching [18] or effects on attentional tracking performance [19]. Furthermore, a compilation of reports for a broad range of substances shows that interactions of serotonin, dopamine, glutamate, and opioid receptors all contribute to aspects of the subjective experience [20]. Other relevant factors for psychedelic action include the relationship between plasma kinetics of a drug, permeability for the blood–brain barrier, duration of target engagement in the central nervous system, and the intensity of psychedelic experience. Finally, the phenomena of “flashbacks” and hallucinogen-persisting perception disorder (HPPD) sometimes occurring in LSD (**1**) users [21] may call for reassessing the notion that visual hallucinations are only due to acute pharmacological activation of serotonin receptors [22].

The first phase of molecular imaging of hallucinogenic compounds employed in vitro binding techniques for assessing affinities and ex vivo methods for revealing the cerebral uptake and binding of radiopharmaceuticals in brain of living animals (e.g., [23]). Molecular imaging by positron emission tomography (PET) and single photon computer tomography (SPECT) have emerged in recent years as mature technologies for monitoring neuroreceptor availability in living brain, for measuring the extent of target engagement by psychoactive drugs [24,25], and to detect physiological responses of the brain to a pharmacological challenge. PET/SPECT methods are admirably suited for studying the cerebral uptake and binding of hallucinogens and for testing effects of psychoactive compounds on physiological markers such as the cerebral metabolic rate for glucose (CMRglc) or cerebral blood flow (CBF). However, this literature is rather sparse; indeed, some well-known hallucinogens remain entirely uninvestigated by molecular imaging techniques.

## 2. Binding Sites of Hallucinogens In Vitro

### 2.1. The Nature of Agonist-Receptor Interactions


Agonism at serotonin receptors is an essential property of hallucinogens. Most serotonin receptors couple to intracellular second messenger systems by one or more guanine nucleotide binding proteins (G-proteins); the presence of guanosine triphosphate (GTP) or its metabolically stable analogues in the receptor binding assay disfavors the binding of agonist ligands, but has no effect on antagonist binding. Thus, the addition of GTP to a binding assay causes a substantial loss of affinity of an agonist ligand in vitro, manifesting in a shift to the right of a displacement curve against the bonding of a labelled antagonist ligand. In general, agonist binding stimulates GTP/GDP exchange, which results in activation of the enzyme adenylate cyclase in the case of the G_s_-type G-protein, inhibition of adenylyl cyclase in the case of G_i/o_, and stimulation of phospholipase C in the case of G_q/11_, among many possible signal transduction pathways. For example, agonists of 5HT_1A_ sites such as 8-hydroxy-DPAT have no intrinsic effect on cyclic AMP (cAMP) production in rat hippocampal neurons, but inhibit the stimulation of adenylyl cyclase provoked by other receptor types [26], thus suggesting receptor coupling to second messenger systems via G_i/o_-type G-proteins. In another assay system, the increased retention of [^35^S]-guanosine-5-*O*-(3-thio)-triphosphate in membranes reveals agonist interactions. Indeed, that assay may serve as a forensic tool for operationally predicting the hallucinogenic properties of members of a series of tryptamine derivatives [27]. However, the mouse head-twitch response behavioral paradigm and rat trials of drug discrimination may serve better to predict hallucinogenic potency of drugs in humans [28].

While the preponderance of evidence indicates that 5HT_2A_ agonism is a necessary property of hallucinogens, this is not sufficient, since certain 5-HT_2A_ receptor agonists such as lisuride (**21**) and ergotamine (**22**) (Figure 2) do not evoke hallucinations (e.g., [29]). Explaining this phenomenon may call for arguments from non-classical pharmacology, whereby agonists can engage different signal transduction pathways through the same receptor. Serotonin 5HT_2A_ receptors typically couple to the Gq/11-mediated signaling pathway, which activates phospholipase C (PLC) to stimulate the formation of inositol phosphates and diacylglycerol, leading to Ca^2+^ release from the endoplasmic reticulum [30]. Although some hallucinogens only weakly stimulate this pathway, hallucinogenic potency may correlate with the efficacy in activating PLC [31]. On the other hand, LSD (**1**) was more effective at activating the 5HT2_A/2C_-coupled phospholipase A2 (PLA2) pathway that mediates arachidonic acid release, whereas the non-hallucinogenic compound 3-trifluoromethylphenyl-piperazine preferentially activated the PLC-IP pathway [32]. Hallucinogenic effects at 5HT_2A_ receptors may also entail activation of the pertussis toxin (PTX)-sensitive heterotrimeric G_i/o_ proteins [33]. As such, it may be the second messenger pathways rather than the particular receptor targets that mediate psychedelic action. Furthermore, the 5HT_2A_ receptor forms a functional heterodimer with the mGluR2 receptor, which evokes allosteric effects on serotonin agonist binding [34]; this interaction reduces the hallucinogen-specific G_i/o_ protein signaling and behavior and may account for the lack of hallucinogenic action of 2-bromo-LSD (**5**) noted above. Certainly, the 5HT_2A_/mGluR2 dimer adds an additional level of complexity to the mechanism of action of hallucinogens.

In the special case of presynaptic autoreceptors, silencing of neuronal electrical activity upon drug application is a functional indicator of agonism. Early electrophysiological research showed that administration of LSD (**1**) at a low dose (5–10 µg/kg) provoked silencing of serotonin neurons in the rat dorsal raphé [35], presumably via activation of 5HT_1A_ autoreceptors. Various other hallucinogenic compounds (psilocin (**8**), *N*,*N*-dimethyltryptamine [DMT (**9**)], and bufotenin (**10**)) inhibit serotonin neuron activity with a potency following the rank order of their potency as hallucinogens [36]. In electrophysiological studies, treatment with 5-HT_1A_ receptor antagonist WAY-100,635 (500 μg/kg, i.v.) prevented the inhibitory effect of LSD (**1**) on the firing rate of dopaminergic ventral tegmental neurons in the rat brain [37]. In the same study, treatment with a blocker of the trace amine-associated receptor (TAAR) type 1 also interfered in the effects of LSD (**1**) on dopamine neuron activity. In vitro binding competition studies indicated that LSD (**1**) and other hallucinogens possess some affinity for TAAR1 [9].

### 2.2. Affinities of LSD at Neuroreceptors In Vitro

LSD (**1**), and likewise mescaline (**6**) (2 mg/kg), and 2,5-dimethoxyphenylisopropylamine (DOM, **11**), silence noradrenergic neurons of the locus coeruleus (but potentiate their response to sensory stimuli) via their effects at serotonin 5HT_2_-like receptors [38]. Thus, to the first approximation, LSD (**1**) and some other hallucinogens act as agonists at serotonin autoreceptors to inhibit serotonin (**43**) release, while also acting at post-synaptic heteroceptors (5HT_2_) to reduce noradrenaline release This action can have the net effect of shifting the bias of serotonin signaling towards the 5HT_2A_ receptors, while blunting noradrenaline signaling. Other research has shown that LSD (**1**) acts as a full or partial agonist at a broad range of serotonin receptors, including 5-HT_1A_, 5-HT_2A_, 5-HT_2B_, 5-HT_2C_, and 5-HT_6_ [39], and has relatively high affinity at dopamine D_3_ receptors in rat striatum cryostat sections [40]. Competition binding studies in vitro indicate that LSD (**1**) has a K_i_ of 3 nM at 5HT_1A_ sites, 4 nM at 5HT_2A_ sites, and 15 nM at 5HT_2C_ sites [9]. In the same study, it was reported that LSD (**1**) has moderate affinity (K_i_) at α_1a_ sites (670 nM) and dopamine D_1_ sites (310 nM) and high affinity at α_2_ adrenergic sites (12 nM), dopamine D_2_ (25 nM) and D_3_ sites (100 nM), but little affinity for monoamine transporters. Others reported K_i_ values for LSD (**1**) of 0.54 nM at 5HT_2A/C_ sites labelled with [^3^H]ketanserin, 0.43 nM at 5HT_1A_ sites, and 6.6 nM at 5HT_1B_ sites [41], thus confirming the considerable 5HT_1_-binding propensity of LSD (**1**).

### 2.3. Affinities of Hallucinogenic Phenylethylamines In Vitro

Under the same assay conditions as with LSD above in [41], the hallucinogenic compounds DOM (**11**), 4-bromo-2,5-dimethoxyphenylisopropylamine (DOB, **12**) and various other phenylpropylamines had nM affinity only at [^3^H]-ketanserin (mainly 5HT_2A_) binding sites in vitro. That series of compounds had 10-fold lower affinity at 5HT_2C_ sites and 1000-fold lower affinity at 5HT_1A_ and 5HT_1B_ sites, indicating much greater selectivity than LSD (**1**). A newer phenethylamine derivative 25CN-NBOH (**13**) had even higher 5HT_2A_ selectivity over 5HT_2C_ and 5HT_2B_ receptors than other reported compounds [42].

Mescaline (3,4,5-trimethoxyphenethylamine, **6**) is a hallucinogenic and psychostimulant natural product occurring in the peyote cactus (*Lophophora williamsii*) and other Mesoamerican cacti, which has been used for millennia for ritual and visionary purposes. Mescaline (**6**) binds with rather low affinity (K_i_) as an agonist at 5HT_1A_ sites (5 µM), 5HT_2A/C_ (6 µM against [^3^H]-ketanserin), alpha_2_ (1.4 µM) and TAAR1 (3.3 µM) receptors [9,43]. Mescaline (**6**) was the first hallucinogen synthesized in the laboratory, and perhaps for this reason found early application in autoradiographic studies [44].

Psilocybin (**2**) ([3-(2-dimethlyaminoethyl)-1*H*-indol-4-yl] dihydrogen phosphate) is a naturally occurring hallucinogen from mushrooms of *Psylocybe* and other genera, which was first isolated in 1959 by the Swiss chemist Albert Hoffman. Initially distributed in the 1960s for use in psychotherapy, psilocybin (**2**) is again finding this application after a hiatus of four decades. Strictly speaking, psilocybin (**2**) is a prodrug, which undergoes rapidly hydrolysis in the gut and liver to yield the active and brain-penetrating metabolite psilocin (**8**). In vitro binding studies indicate that psilocin (**8**) is an agonist with moderate affinity (K_i_) of 49 nM at 5HT_2A_ sites, 94 nM at 5HT_2C_ sites and 123 nM at 5HT_1A_ sites, but >µM affinity at dopamine and adrenergic receptors [9]. Screening indicated a moderate affinity for psilocin (**2**) at 5HT_2B_ sites [45], which may have some bearing on the propensity of psilocybin to cause cardiac valvulopathy. The corresponding Ki values against [^123^I]-DOI indicated ten-fold higher affinity at 5HT_2A_ sites [10,14], which is presumably indicative of the greater competition between an agonist drug at agonist-labelled receptors, as has been established for the case of dopamine D_2/3_ receptors [46].

### 2.4. Affinities of Hallucinogenic Tryptamines In Vitro

The prototypic tryptamine hallucinogen DMT (**9**) had a K_i_ of 0.5 µM [47] or 2 µM [48] against (mainly) 5HT_2A/C_ sites labelled with [^3^H]-ketanserin, indicating rather low affinity in vitro. Other displacement studies against [^3^H]-ketanserin likewise showed µM Ki at 5HT_2A/C_ sites and insensitivity to addition of GTP, suggesting an antagonist interaction [49]. *Prima facia*, this seems unlikely since DMT (**9**) substituted for a 5HT_2A_ agonist in a rat behavioral drug discrimination trial against the antagonist ketanserin (**7**) [50]. Furthermore, pretreatment with ketanserin (40 mg) blocked the visual hallucinations and electroencephalographic effects otherwise evoked by DMT (**9**) in healthy volunteers [51]. The generally low affinity of DMT (**15**) seen in [^3^H]-ketanserin displacement studies in vitro seems at odds with its much greater potency in functional assays, which showed a 38 nM EC_50_ for calcium mobilization in vitro [52]. This discrepancy doubtless reflects the ten-fold better competition of a series of indolylalkylamines such as 5-MeO-DMT (**15**) against [^123^I]-DOI (**16**)-labelled 5HT_2A_ agonist sites as compared to [^3^H]-ketanserin-labelled 5HT_2A_ antagonist sites [53]. On the other hand, in the displacement study [49], DMT had a 100 nM Ki against 5HT_1A_ sites labeled with the agonist [^3^H]8-hydroxy-DPAT and a clear shift to the right in the displacement curve in the presence of GTP, indicating agonism. Others reported that DMT (**9**) had K_i_s of 0.25 µM at 5HT_2A_ sites, 0.4 µM at 5HT_2C_ sites, and only 0.1 µM at 5HT_1A_ sites [9], again suggesting that DMT (**9**) may have a preferred action at 5HT_1A_ sites. DMT (**9**) also showed an interaction with moderate affinity at sigma receptors [54]. Indeed, the activity of DMT (**9**) at sigma-1 receptors mediates cytoprotective properties against hypoxic neuron injury in vitro, and in the reperfusion injury model in vivo [55].

5-methoxy-*N*-methyl-*N*-isopropyltryptamine (5-MeO-MiPT, **14**) is a hallucinogenic substituted tryptamine with moderate affinity at 5HT_1A_ receptors (Ki = 60 nM) and 5HT_2A_ (receptors K_i_ = 160 nM) [9]. 5-MeO-DMT (**15**) has similar affinity at 5HT_2A_ receptors (Ki = 200 nM), under binding conditions where LSD (**1**) had a K_i_ of only 1 nM [47], and had >100-fold selectivity for 5HT_1A/B/C_ receptors relative to 5HT_2A_ receptors in vitro [56]. Thus, 5HT_1A_ selectivity may be a general property of tryptamine hallucinogens. The 5-MeO-DMT (**15**) active metabolite 5-hydroxy-DMT (5-HO-DMT; bufotenine (**10**)) is a hallucinogenic compound present at psychoactive doses in the skin of the Colorado river toad *Incilius alvarius,* and other toad species. Bufotenine (**10**) had a Ki of 14 nM against 5HT_2A_ binding sites labelled with the agonist [^125^I]-*R*-(-)DOI (**16**) [10]. Despite its relatively low affinity in vitro [56], 5-MeO-DMT (**15**) had a comparably high EC_50_ of 4 nM at 5HT_2A_ sites in a functional assay of intracellular calcium mobilization [52], confirming its higher potency at these sites compared to DMT (**9**). Indeed, the strong association between hallucinogenic properties of a broad range of structural classes of molecules with 5HT_2A_ affinity supports the use of antagonist drugs like ketanserin (**7**) as a treatment for drug-induced hallucinations [47]. On the other hand, pretreatment with the 5HT_1A_ antagonist pindolol potentiated the hallucinogenic effect of intravenous DMT (**9**) in healthy volunteers, suggesting a breaking effect of 5HT_1A_ receptors on 5HT_2A_-evoked hallucinations in humans [57]. Furthermore, the IC_50_ of DMT (**9**) and its derivatives at 5HT_2A_ sites did not seem to correlate with visual hallucination intensity in humans [52].

### 2.5. The Strange Case of Ibogaine

Ibogaine (**17**) is a psychoactive compound from root bark of the iboga tree (*Tabernanthe iboga*), which is known as an oneirogen for the dreamlike quality of the hallucinations it can provoke. The characteristics of the ibogaine (**17**) structure are an indole, a tetrahydroazepine, and a bicyclic isoquinouclidine. An anecdotal report that ibogaine (**17**) reduced opioid craving found support from rat studies showing an attenuation of the morphine-induced dopamine release in rat striatum [58]. By some accounts, ibogaine (**17**) can protect from relapse from a wide range of abused substances of distinct pharmacology. Unlike other hallucinogens discussed in this review, ibogaine (**17**) was without notable affinity for serotonin 5HT_2A_ receptors. However, its metabolite noribogaine (**18**) possessed some affinity (Ki 2 µM) as a partial agonist for µ-opioid [59] and κ-opioid receptors [60], although much less so than the κ-opioid agonist hallucinogen Salvinorum A. Ibogaine (**17**) also had some affinity for NMDA receptors, displacing [^3^H]MK801 from caudate membranes with an IC_50_ of 5 µM [61], which may account for its dissociative side effects. Ibogaine (**17**) also had moderately high affinity (Ki 2 µM) at dopamine transporters [62], albeit by binding to their inward facing conformation, which may attenuate amphetamine-evoked dopamine release. Ibogaine (**17**) and its metabolite noribogaine (**18**) enhanced the G_i/o_-mediated inhibition of adenylyl cyclase by morphine or serotonin, while having no intrinsic effect on basal or forskolin-stimulated adenylyl cyclase [63]. This phenomenon may account for ibogaine’s (**17**) putative efficacy in treating addiction.

Ibogaine (**17**) has a considerable degree of toxicity, which has led to a number of fatalities. The less toxic ibogaine (**17**) congener 18-methoxycoronaridine (**19**) is likewise a putative anti-addictive compound, apparently due to its antagonism at α_3_β_4_ nicotinic acetylcholine receptors [64]. Tabernanthalog (**20**) and a series of other ibogaine (**17**) analogues lacking the isoquinuclidine retained the capacity to promote synaptic plasticity, while being much less cardiotoxic and teratogenic than ibogaine (**17**) itself [65].

## 3. Ex Vivo/In Vitro Binding Studies with Hallucinogens


### 3.1. LSD Derivatives


Early research into structure-function indicated that methylation of ergotamines at the *N*_1_ position enhances serotonin antagonism in the isolated rat uterus assay [66], while decreasing hallucinogenic potency predicted from a quantitative structure-activity relationship (QSAR) study [67].*N*-methyl-2-[^125^I]-iodo-lysergic acid diethylamide ([^125^I]-MIL, **24**) was developed as a presumably non-hallucinogenic ligand for the molecular imaging of serotonin receptors, whereby *N_-_*methylation of [^125^I]-LSD (**25**) was intended to impart greater selectivity and sensitivity for 5HT_2_ receptors [23]. Studies in vitro with rat forebrain homogenates indicated that [^125^I]-MIL (**24**) had an apparent K_D_ of 0.14 nM at 5HT_2A_ receptors. However, [^125^I]-MIL (**24**) also showed a K_D_ of 0.4 nM for 5HT_2C_ receptors in vitro. The specific binding of [^125^I]-MIL (**24**) in mouse brain peaked at 45 min post injection, when the binding ratio relative to cerebellum was 4 in frontal cortex and 2 in striatum; lesser cortical binding and more rapid washout from frontal cortex was seen in corresponding studies with [^125^I]-LSD (**25**). In other ex vivo studies, ketanserin (**7**) evoked a dose-dependent displacement of [^125^I]-MIL (**24**) in mouse striatum and cortex, suggesting an IC_50_ of about 30 µg/kg ketanserin (**7**). Rat autoradiographic studies with [^125^I]-MIL (**24**) revealed that repeated doses with the non-hallucinogenic psychostimulant MDMA evoked a substantial down-regulation of 5HT_2_-like receptors [68], which may be a marker of the phenomenon of tolerance to certain hallucinogens.

*D*-(+)-*N*-ethyl-2-[^125^I]iodo-lysergic acid diethylamide ([^125^I]-EIL, **23**) was developed as a ligand for molecular imaging of serotonin receptors. It had very high affinity for 5HT_2A_ receptors in rat cerebral cortex, with a dissociation constant (K_D_) of 0.2 nM [69]. Following the precedent of N_1_-methylation, we suppose that [^125^I]-EIL (**23**) is likely to be an antagonist. Ex vivo studies indicated an extraordinary persistence of its specific binding in mouse brain relative to cerebellum, whereby the binding ratio was 9 at 16 h post injection. Ketanserin (**7**) displaced the cerebral binding, but dopamine D_2_ or adrenergic antagonists were without effect, consistent with a main interaction of [^125^I]-EIL (**23**) at 5HT_2A_ sites. However, the authors predicted that [^125^I]-EIL (**23**) might also bind to 5HT_2C_ receptors of the choroid plexus.

The active *D*-enantiomer of LSD (**1**) had 1000-fold higher affinity for serotonin receptors than the *L*-enantiomer [70]. Autoradiographic studies with *D*-[^125^I]-LSD (**25**) (200 pM) showed abundant binding in the extended striatum and the cerebral cortex, and nearly complete displacement of the cortical binding be co-incubation with *R*-(-)-DOB ((-)**12**, 500 pM), but only 50% displacement in striatum, consistent with the ambivalence of LSD (**1**) for dopamine and serotonin receptors [71]. Other autoradiographic studies with *R*-[^125^I]-DOI (**16**) showed an abundance of LSD-displaceable binding in the deep layers of the cerebral cortex and in the claustrum. However, there was only sparse binding in striatum, consistent with the ligand’s considerable specificity for serotonin receptors [72]. More detailed autoradiographic examination of *D*-[^125^I]-LSD (**25**) binding in rat brain indicated a single population of binding sites (K_D_ 170 pM) in cerebral cortex, where the B_max_ was about 4 pmol/g wet weight [73]. The binding in striatum was of similar density, but with a higher apparent K_D_ (300 pM), indicative of the slightly lower affinity of LSD (**1**) for dopamine D_2_-like receptors. However, other binding studies with [^125^I]-LSD (**25**) revealed a significant ketanserin (**7**) displaceable component in rat striatum sections, ranging from 30% in rostral parts to 74% in the caudal regions [74].

### 3.2. Phenylethylamine Derivatives

Autoradiographic analysis of the rat brain revealed the time-dependent distribution of radioactivity at various times after intravenous injection of [^14^C]-psilocin at a dose of 10 mg/kg [75]. There was rapid initial cerebral uptake, resulting in concentrations of approximately 1% ID/g (injected dose/gram) at one-minute post injection. At 60 and 120 min post injection, binding remained high in the anterior cingulate cortex, amygdala, and hippocampus, with relatively lower concentrations in white matter. There was substantial washout of radioactivity from brain between four and eight hours post injection. This pharmacokinetic analysis was of total brain radioactivity, uncorrected for possible brain-penetrating metabolites, or metabolism in brain.

Studies with α-[^14^C]-mescaline in cat (25 mg/kg) showed a plasma half-life of two hours after intravenous injection, very rapid uptake into brain giving a brain/plasma ratio of 3 at one hour after injection, and persistent retention in brain, with only slight washout at six hours. Autoradiographic studies with [^3^H]-mescaline in the brain of marmoset monkeys (*Callithrex jacchus*) showed preferential accumulation in the hippocampus, amygdala, lateral geniculate and anterior cingulate cortex, persisting even 18 h after administration (8 mg, i.p.) [76]. Similar studies in mice with the non-hallucinogenic isomer of mescaline, [^3^H]-2,3,4-trimethoxy-β-phenylethylamine (4 mg) showed a more homogeneous pattern of binding [77], suggesting that mescaline (**6**) evokes hallucinations through specific receptors or binding sites. However, sensitive analyses with GS-MS showed a brain:serum ratio of only 0.3 at 60 min after mescaline (**6**) administration in rats (20 mg/kg, s.c.), with slow washout from brain suggesting a three-hour half-life [78]. Rats showed disruption of the startle reflect at 60 min but not at 10 min after treatment with mescaline (**6**) at the 10 mg/kg dose (s.c.), which presumably reflects the delay to absorption and brain entry.

2,5-Dimethoxy-4-iodoamphetamine (DOI) is a prototypical phenylethylamine 5-HT_2A/2C_ receptor agonist, although having some affinity for 5HT_1A_ receptors, as does a wide range of its phenylalkylamine derivatives [79]. Autoradiography with [^125^I]-DOI (**16**) can be used to study the functional desensitization of serotonin 5HT_2_-likereceptors after repeated doses of DOI or other agonists. Thus, chronic treatment with (+/−)-DOI (1 mg/kg daily for a week) significantly reduced the binding of [^3^H]ketanserin, [^125^I]-LSD (**25**), and *R*-[^125^I]-DOI (**16**) as measured at single ligand concentrations in rat cortical homogenates [80]. Saturation binding studies indicated that chronic DOI treatment significantly lowered the B_max_ values for [^3^H]-ketanserin and *R*-[^125^I]-DOI (**16**) without altering the K_D_ values. Repeated treatment of rats with (+/−)-DOI (1 mg/kg, i.p.) resulted in an attenuation of the DOI-induced release of the hypothalamic hormone oxytocin, which is indicative of receptor desensitization [81]. This functional desensitization was associated with reduced autoradiographic binding of *R*-[^125^I]-DOI (**16**) in the paraventricular nucleus, despite increased 5HT_2A_ immunoreactivity to western blot analysis, suggesting an internalization process or altered coupling with intracellular G-proteins.

The compounds 2-(4-iodo-2,5-dimethoxyphenyl)-*N*-(2-methoxybenzyl)ethanamine (25I-NBOMe, **31**) and 2-[[2-(4-iodo-2,5dimethoxyphenyl) ethylamino] methyl] phenol (25I-NBOH, **32**) (Figure 3) are hallucinogenic 5HT_2A_ agonists that have gained considerable notoriety due to cases of fatal intoxication. They undergo hydroxylation, O-demethylation, N-dealkylation, and dehydrogenation in vivo, mainly via CYP3A4 and CYP2D6 [82]. Relative to 2,5-dimethoxyphenethylamines (2C compounds) and DOI, the *N*-methoxybenzylated compounds had much higher affinity for 5-HT_2A_ receptors labelled with the antagonist ligand [^3^H]-MDL100907 in vitro [83]. Binding assays against [^3^H]-ketanserin/[^3^H]-mesulergine, [^3^H]-LSD and [^3^H]-Cimbi-36 in vitro showed that 25CN-NBOH (**13**, *N*-(2-hydroxybenzyl)-2,5-dimethoxy-4-cyanophenylethylamine) had >52-fold *K*_i_^2C^/*K*_i_^2A^ selectivity ratio and a 37-fold *K*_i_^2B^/*K*_i_^2A^ ratio, and likewise showed substantial 5HT_2A_ selectivity in functional assays of IP turnover. Various isomers of 25H-NBOMe in which the two methoxy-groups are in different positions (ortho, meta, and para) were compared with respect to their efficacy in activating signaling pathways, in conjunction with molecular docking studies at the 5HT_2A_ receptor binding pocket [66]. In the series examined, several dimethoxy compounds proved to have an efficacy exceeding that of LSD (**1**).

## 4. Molecular Imaging Studies In Vivo with Hallucinogens


### 4.1. LSD Derivatives

One of the very first receptor PET molecular imaging studies employed N1-([^11^C]-methyl)-2-bromo-LSD ([^11^C]-MBL, **33**) as a non-hallucinogenic ligand for serotonin 5-HT_2A_ receptors [84]. The authors first tested the pharmacological specificity of MBL in vitro, finding a K_i_ of 0.5 nM at 5HT_2_ receptors, versus 4 nM at D_2_ receptors and weaker interactions with α_1_ adrenergic (K_i_ = 250 nM) and serotonin 5HT_1A_ receptors (K_i_ = 34 nM). A PET study with [^11^C]-MBL (**33**) in baboon (0.1 µg/kg body weight) indicated relatively high binding throughout cerebral cortex, with a peak binding ratio of 1.4 at 40 min that tended to decline in the following 45 min; this binding was displaced by ketanserin (**7**) pretreatment. Similar results were seen in an awake human, with a maximal binding ratio of about 2:1 prevailing in cerebral cortex during 20–60 min post-injection.

### 4.2. Phenylethylamine Derivatives

In a SPECT study of healthy human volunteers, oral administration of 4-bromo-2,5-dimethoxy-phenylisopropylamine (2,5-dimethoxy-4-bromoamphetamine; DOB, **12**) labelled with ^77^Br or ^82^Br showed a rather delayed cerebral uptake in brain of human volunteers, which the authors attributed to a metabolite rather than the parent compound [85]. Subsequent SPECT studies with 2,5-dimethoxy-4-[^123^I]-iodoamphetamine (*R-*[^123^I]‑DOI; [1-(4_iodo-2,5-dimethoxyphenyl)-propane-2-amin]) of high specific activity showed high uptake in thalamus, cortex, and striatum, and lesser accumulation in the midbrain and cerebellum in baboon [86]. However, there was no displacement of the uptake by treatment with ketanserin (**7**), suggesting extensive entry of labelled metabolites into brain.

The phenylethylamine 5-HT_2A_ receptor agonist 2-(4-iodo-2,5-dimethoxyphenyl)- *N*-(2-[^11^C-OCH_3_]methoxybenzyl) ethanamine ([^11^C]-Cimbi-5, **34**) was investigated as a potential PET tracer in rats ex vivo and in pigs in vivo [87]. It had a Ki of 2 nM at 5HT_2A_ and 5HT_2B_ sites, 7 nM at 5HT_2C_ sites, and much less affinity at other neuroreceptors in vitro. Rat-PET studies showed a peak binding ratio at about 30 min after tracer injection, and nearly complete displacement by ketanserin (**7**) treatment, and pig PET studies showed 75% displacement by ketanserin (**7**). HPLC analysis of plasma samples indicated that a lipophilic metabolite came to comprise about 20% of the plasma radioactivity; this metabolite was nonetheless absent from brain and thus unlikely to interfere with reference tissue quantitation of [^11^C]-Cimbi-5 (**34**) binding. Others conducted a comparative PET study of [^11^C]-Cimbi-5 (**34**) and the antagonist [^11^C]-MDL100907 (**35**) in living non-human primates, finding that the antagonist ligand had two or three-fold higher BP_ND_ than [^11^C]-Cimbi-5 (**34**) in most brain regions [88]; pretreatment with MDL100907 did not completely displace [^11^C]-Cimbi-5 (**34**) binding in all brain regions, indicating somewhat incomplete selectivity.

A series of phenylethyalamines in the Cimbi series showed Kis in vitro against the binding of [^3^H]-MDL100907 ranging from 0.2 nM (Cimbi-31, **36**) to 47 nM (Cimbi-88, **37**); their receptor agonism was assessed from PI hydrolysis in GF62 cells overexpressing the 5-HT_2A_ receptor [89]. Furthermore, a series of nine structurally similar compounds of the Cimbi phenethylamine class had a ten-fold range in the affinity at 5HT_2A_ sites and a 100-fold range in their ED_50_ for activation of IP hydrolysis [89]. However, neither property predicted the specific binding (BP_ND_) in brain, which ranged from 0.2 to 1.0. The best of the series with respect to specific signal and washout from cerebellum was [^11^C]-Cimbi-36 (**38**). In brain of living rat, [^11^C]-Cimbi-36 shows preferential uptake in frontal cortical regions and relatively little uptake in the striatum, diencephalon and cerebellum (Figure 4).

Further PET investigations of [^11^C]-Cimbi-36 (**38**) in healthy volunteers showed BP_ND_s, which ranged from 0.25 in striatum to 1.7 in broad cortical regions, and showing a high degree of covariance between repeat measures in the same subjects [90]. In a head-to-head study there was generally high correlation between [^11^C]-Cimbi-36 (**38**) and [^18^F]-altanserin (**39**) BP_ND_ estimates in most brain regions, but significantly greater binding of [^11^C]-Cimbi-36 (**38**) in hippocampus and choroid plexus. The authors interpreted this to indicate the composite of 5HT_2A_ and 5HT_2C_ receptors in that structure; whereas [^11^C]-Cimbi-36 is nearly equipotent at 5HT_2A_ and 5HT_2C_ receptors, altanserin has nearly 100-fold selectivity for 5HT_2A_ binding sites [91]. Similar studies in non-human primate supported those findings, showing a high degree of concord between [^11^C]-Cimbi-36 (**38**) and [^11^C]-MDL100907 (**35**) binding in most brain areas, and substantial displaceability by ketanserin (**7**) in all regions but cerebellum [92]. However, there was only partial displaceability of [^11^C]-Cimbi-36 (**38**) binding in hippocampus and choroid plexus by treatment with the 5HT_2C_ antagonist SB 242084 (**40**), again consistent with a mixed signal in those regions. The metabolism of [^11^C]-Cimbi-36 (**38**) in humans is characterized by loss of the [^11^C]-methoxy label (generating [^11^C]-formaldehyde) or demethylation of the other methoxygroup, yielding a relatively hydrophobic radiometabolite [93].

### 4.3. Tryptamine Derivatives

DMT (**9**) obtained a brain:blood partition ratio of about 5:1 after intraperitoneal injection in the rat, but disappears rapidly from tissue and circulation due to metabolism [94]. A planar scintigraphy study examined the biodistribution of radioactivity in rabbits following intravenous injection of 2-[^131^I]iodo-DMT [95]. The brain activity reached a peak about 30 s after injection, when the brain:blood ratio was approximately unity in the olfactory bulb, thus indicating very fast passage across the blood–brain barrier. Remarkably, traces of activity remained in the olfactory bulb (but not elsewhere in the brain) for several days after administration; the authors suggested that this might reflect trapping in vesicles and noted that a rabbit study with 2-[^131^I]-iodo-tryptamine showed no comparable retention in brain. They authors did not undertake any displacement studies to confirm specific binding, nor did they pretreat the rabbits with a monoamine oxidase (MAO) inhibitor, arguing that the intravenous route of administration avoided first pass metabolism in the liver.

### 4.4. Competition from Hallucinogens at Dopamine Receptors In Vivo

In the PET competition paradigm, reductions in the availability of sites for radioligand binding can reveal occupancy or target engagement after challenge with another pharmaceutical that interacts with the same target. Alternatively, a treatment that mobilizes the release on an endogenous neurotransmitter can also evoke displacement of the radiopharmaceutical in living brain. This is a well-established paradigm for detecting competition at dopamine D_2/3_ receptors in living striatum. In one such PET study, healthy volunteers were scanned with the D_2/3_ antagonist [^11^C]-raclopride (**41**), first in a placebo baseline, and again after administering a psychoactive dose of psilocybin (**2**) (0.25 mg/kg p.o.) [96]. Comparison of the two scans revealed a global 20% decline in [^11^C]-raclopride (**41**) binding in the striatum, which is similar to the displacements evoked by amphetamine or other strong releasers of dopamine. Furthermore, the decline in the ventral striatum correlated with individual ratings for euphoria and depersonalization. In formal terms, the reduced [^11^C]-raclopride (**41**) binding could be indicative of competition from psilocin (**8**) at the D_2/3_ receptors, or enhanced release of endogenous dopamine. Rat microdialysis studies showed that psilocin (**8**) (10 mg/kg) evoked a transient 40% increase in interstitial dopamine levels [97], which would seem insufficient to account for the 20% displacement by low dose psilocybin (**2**) in humans. Nor does the reported affinity of psilocin (**8**) at dopamine receptors seem sufficient to support a significant direct occupancy in vivo.

The effects of intravenous challenge with acute LSD (**1**) (2.5 µg/kg) on dopamine D_2/3_ receptor availability was tested in a [^11^C]-raclopride (**41**) PET study of anesthetized pigs [98]. There was no immediate effect on the striatal [^11^C]-raclopride (**41**) BP_ND_, but there was a 19% decline at four hours after the challenge, which seemed consistent with a high affinity component of the displacement of [^3^H]-raclopride binding by LSD (**1**) in pig brain sections. Furthermore, their rat microdialysis studies did not indicate any effect of LSD (**1**) on dopamine release per se, suggesting that the [^11^C]-raclopride (**41**) binding reduction must be due to direct competition from LSD. However, the time delay to a main effect suggests that some other pharmacological mechanism or methodological factor may have been at play. The pharmacokinetics of oral LSD (**1**) (100 and 200 µg) in human subjects indicated a mean plasma half-life of 2–6 h and a persistently close relationship between the individual LSD (**1**) concentration in plasma and the magnitude of subjective response within subjects, with moderate counterclockwise hysteresis during the 12 h after dosing [99]. Thus, one might have expected simple competition by LSD (**1**) at [^11^C]-raclopride binding sites in pig brain in the first post LSD (**1**) scan, rather than with a delay of several hours.

Further PET studies of a generalization of the ayahuasca phenomenon (see below) were intended to test the hypothesis that MAO inhibition should potentiate the amphetamine-evoked release of dopamine in living striatum. Here, we used the amphetamine challenge paradigm to detect increased competition by dopamine at striatal dopamine D_2/3_ receptors labelled by [^11^C]-raclopride. The amphetamine challenge evoked the expected 15% displacement of [^11^C]-raclopride in striatum of living pigs, but there was no potentiation of the amphetamine effect in pigs pretreated with pargyline at a dose sufficient to block completely the specific binding of [^11^C]-harmine (**48**) (Figure 5) to MAO-A [100]. Neither was the expected potentiation of amphetamine-evoked dopamine release seen in a [^11^C]-raclopride PET study in rats with MAO inhibition [101]. Conceivably, increased signaling by tyramine or phenylethylamine at TAARs upon the treatment with an MAO inhibitor may have interfered via autoreceptor effects with the potentiation of dopamine release [102]. We have developed [^18^F]-fluoroethylharmol (**49**) as an alternative PET tracer for MAO-A in brain [103], with potential applications in the study of the pharmahuasca phenomenon.

### 4.5. Competition from Hallucinogens at Serotonin Receptors In Vivo


In a generalization of the competition PET model, reductions in the availability of binding sites for the serotonin 5HT_2A_ antagonist [^18^F]-altanserin (**39**) are indicative of increased release of endogenous serotonin after challenge with dexfenfluramine [104], and/or target engagement by competing drugs. In a PET study of 11 healthy young male subjects, [^18^F]-altanserin (**39**) binding was measured at baseline, and (two weeks later) again at 75 min after administration of psilocybin (**2**) (20 mg). The drug treatment provoked widespread 50% decreases in the uptake (DV’) of the tracer in the dorsolateral prefrontal cortex, orbitofrontal cortex, medial temporal cortex, and other cortical regions. Voxelwise regression analysis showed that the binding reductions in the anterior cingulate spreading to the dorsomedial prefrontal cortex correlated with higher scores in the Altered States of Consciousness Rating Scale (5D-ASC) (Figure 6).

In an exemplary study of its type, Madsen et al. tested the occupancy of psilocin (**8**) at serotonin 5HT_2A_ receptors in brain of human volunteers relative to plasma drug concentrations after administration of psilocybin (**2**) at doses in the range of 3 to 30 mg [106]. In the Madsen PET study, cerebral 5HT_2A_ receptor availability was measured using the agonist ligand [^11^C]-Cimbi-36 (**38**). Results indicated a dose-dependent brain receptor occupancy as high as 76%, which occurred when the plasma psilocin (**8**) concentration was about 100 nM. Furthermore, the intensity of the psychedelic experience correlated with receptor occupancy, and with the plasma psilocin (**8**) concentration, assuming a single site model. These calculations assumed steady states in the drug concentration and intensity during the PET measurement, which the authors showed to be approximately true on the time scale of about two hours. Their data also showed a rough relationship between the duration of the hallucinogenic experience (60–360 min) and the interval in which the plasma psilocin (**8**) concentration exceeding approximately 10 nM. Interestingly, the occupancy by psilocin (**8**) at [^11^C]-Cimbi-36 (**38**) agonist binding sites (circa 60%) was quite similar to the displacement of [^18^F]-altanserin (**39**) by a similar dose of psilocybin (**2**) reported by Hasler et al. [105]. This finding calls into question the assumption that agonist receptor ligands are intrinsically fitter than antagonist ligands for the detection of competition from other agonists. Nonetheless, Jorgensen et al. argued that the occupancy of 5HT_2A_ sites labelled with [^11^C]-Cimbi-36 (**38**) by endogenous serotonin in anesthetized pig increased from 17% at baseline to 44% after fenfluramine treatment, which transiently increased the interstitial serotonin content in cortex ten-fold according to microdialysis [104]. This sensitivity of [^11^C]-Cimbi-36 (**38**) binding to fenfluramine challenge seemed to exceed that reported earlier for [^18^F]-altanersin (**39**) in healthy human volunteers, albeit at a lower relatively lower dose of fenfluramine in human [104].

At baseline in healthy individuals, at least, there was no significant association between the trait for openness to experience with 5HT_2A_ availability, either measured with the antagonist [^18^F]altanserin (**39**), or with the agonist ([^11^C]-Cimbi-36 (**38**) [107]. However, a single dose of psilocybin (**2**) provoked in healthy individuals a long-lasting increase in scores for the personality traits of openness to experience and mindfulness, which correlated inversely with reductions in [^11^C]-Cimbi-36 (**38**) binding [106]. Recent work shows that a single dose of psilocybin (**2**) at a presumably hallucinogenic dose (0.1 mg/kg, i.v.) reduced by up to 50% the cortical abundance of 5HT_2A_ binding sites labelled in vitro with [^3^H]-MDL100907 or [^3^H]-Cimbi-36 [108]. Other examinations in the same pig brain samples with the synaptic vesicle protein 2A (SV2A) ligand [^3^H]-UCB-J showed a significant 5–10% *increase* in synaptic density in hippocampus, occurring in association with the down-regulation of serotonin receptors. This phenomenon suggests that the benefits of psilocybin (**2**) in a therapeutic setting may be due to an enhancement of synaptic density and or plasticity; it might be more correct to suggest that the treatment reinstates the synaptic overshoot that occurs early in post-natal brain development, which may set the stage for plastic changes in synaptic plasticity.

In a recent rodent study, a single dose of DOI exerted rapid effects on the dendritic spine structure in frontal cortex that were dependent on 5HT_2A_ receptors [109]. The same treatment evoked chromatin changes that persisted long after the DOI treatment, suggesting that therapeutic responses to hallucinogen treatment may arise through persistent and epigenetic changes in synaptic architecture. Others have recently used chronic two-photon microscopy to monitor synaptic spine density in mouse cortex [110]. In female mice, a single dose of psilocybin (1 mg/kg, i.p.) evoked a 20% increase spine density in layer V of the frontal cortex that persisted for up to 34 days, along with a sharp increase in spine dimension that peaked at day one and declined over the following week; these changes were scarcely evident in male mice. Overall, these structural studies concur with the SV2A results discussed above, and fit into a general framework whereby rapid antidepressant responses to ketamine also occur along with dendritic expansion, albeit by a different mechanism [111].

## 5. Metabolism of Hallucinogenic Compounds

### 5.1. LSD

LSD (**1**, 100 µg) attained a maximal plasma concentration of 4 nM at two hours after oral administration in healthy volunteers, and declined thereafter with an approximately four hour half-life [99]. The subjective effects, both good and bad, closely tracked the plasma concentrations, as did physiological responses, such as heart rate and blood pressure increases. The metabolism of LSD (**1**) is complex, but the predominant pathway in human involves microsomal formation of 2-oxo-3-hydroxy-LSD; whereas N-demethylation occurs in animals, this may not be an important pathway in humans [112].

### 5.2. Phenylethylamines

After intravenous of [^14^C]-psilocin in the rat, there was rapid uptake of radioactivity in the brain, followed by substantial biliary elimination [113]. Chromatographic analysis indicated that the major eliminated metabolites were the glucoronate and 4-hydroxyindoleacetic acid. In rats, metabolism of psilocin (**8**) occurs in part by oxidative deamination to 4-hydroxy-3-indoleacetic acid, but in humans, the predominant metabolite is psilocin-glucuronate [114]. As noted above, psilocybin (**2**) undergoes rapid de-phosphorylation after oral administration, and the centrally active metabolite psilocin (**8**) had a plasma elimination half-life of about three hours in healthy humans [115].

The diemethoxyphenethylamines such as [^11^C]-Cimbi-36 are metabolized in pigs and humans by demethylation, usually in the 5-position, and subsequently glucuronation [116]. Comparison of two forms of [^11^C]-Cimbi-36 showed that labelling on the 5-position yielded a lower specific binding in pig brain, apparently due to the formation of [^11^C]-formaldehyde or other brain-penetrating labelled metabolites [117]. A recent detailed pharmacokinetic analysis of the 5HT_2A_-prefering agonist 25CN-NBOH showed a complex metabolic profile in living rats, with debenzylation, demethylation, and hydroxylation, mediated by perhaps half a dozen cytochrome enzymes; the compound showed good distribution across the blood–brain barrier, measured ex vivo [118].

### 5.3. Tryptamimes

Remarkably, there is evidence that DMT (**9**) may be an endogenous neurotransmitter. Ultracentrifugation studies of rat brain indicated that DMT (**9**) is present in a neuronal vesicle fraction, along with its precursor, tryptamine (**42**) [119]. The enzyme indolethylamine-N-methyltransferase (INMT) is present in mammalian brain [120] and can catalyze the formation of DMT (**9**) from tryptamine (**42**), or bufotenin (**10**) from serotonin [121]. The brain enzyme uses S-adenosylmethionine as a methyl donor for the successive formation of *N*-methyltryptamine and then DMT (**9**) [122]. The INMT enzyme co-localizes in ventral horn cholinergic synapses called C-boutons/C-terminals expressing the sigma1 receptor, which was previously implicated in the neuroprotective action of DMT (**9**) [123]. Cerebral microdialysis studies in rats indicated a cortical interstitial DMT (**9**) concentration of about 1 nM, which was similar to that measured for serotonin in the same samples. The cortical DMT (**9**) measurements were unaffected by removal of the pineal gland, which has the highest INMT activity and DMT (**9**) concentration in the body, thus indicating local formation in brain [124]. Indeed, the production of DMT (**9**) by the pineal gland would hardly suffice to provoke central effects via its release into the circulation and subsequent uptake across the blood–brain barrier [125], which may cast doubt upon the plausibility of a physiological role for DMT (**9**). The substituted tryptamine compounds are characterized by rapid metabolism in vivo through O-demethylation by CYP2D6, and by oxidative deamination by MAO [126], such that the main metabolites of 5-MeO-DMT (**15**) in mouse urine are DMT-glucuronate and 5-methoxy-indoleacetic acid [127]. As shall be discussed below in some detail, the importance of MAO-A for the first past metabolism of DMT (**9**) accounts for the Ayahuasca effect, in which the hallucinogenic effects of oral DMT (**9**) are potentiated by co-treatment with an MAO inhibitor. This may have some bearing on the possible physiological actions of endogenous DMT (**9**), at least under conditions of pharmacological MAO inhibition.

## 6. Ayahuasca and Pharmahuasca

As noted above, DMT (**9**) undergoes rapidly metabolism by MAO in the gut and liver, and does not normally attain sufficient blood levels to evoke a hallucinogenic experience, unless taken at very high doses. Ayahuasca refers to a traditional method to enhance the effectiveness of DMT (**9**) by co-treatment with an MAO inhibitor. Ayahuasca is a Quechua word from *Aya* meaning “spirit, soul” in reference to its visionary effects and *Waska*, meaning “rope” or “woody vine”, referring to its source from the vine *Banisteriopsis caapi* together with the leaves of the shrub *Psychotria viridis*. The latter plant contains DMT (**9**), a hallucinogenic tryptamine that acts as a 5-HT_1A/2A/2C_ agonist [54,128]. However, *Banisteriopsis caapi* supplies various β-carboline alkaloids (harmine (**44**), tetrahydroharmine (**45**), and harmaline (**46**)) that act as reversible inhibitors of MAO type A (IC_50_ 2 nM [129]). Thus, the consumption of the two plants together results in a potentiation of the absorption of DMT (**9**), which provokes a hallucinogenic experience lasting 2–4 h. Practiced since pre-Columbian times, Ayauhuasca retains an important ritual function among certain Amerindians, and Brazilian law protects its use as such. Ayahuasca, or its chemical equivalent known as pharmahuasca, has emerged as a tool for psychotherapy in the contexts of affective and substance use disorders [130,131], thus calling for a better understanding of its possible mechanisms of action.

There is abundance of preclinical evidence indicating that MAO inhibition potentiates the central action hallucinogenic tryptamines. For example, treatment of rats with harmaline (**46**) (2–15 mg/kg, i.p.) greatly potentiated the behavioral effects of 5-MeO-DMT (**15)**, causing hyperactivity at a subthreshold dose of the tryptamine compound (2 mg/kg, i.p.) [132]. Furthermore, 5-MeO-DMT (**15**) (1 mg/kg i.p.) was without effect on pre-pulse inhibition of the startle reflex in rats, unless following upon treatment with an MAO-A inhibitor; the pretreatment increased the plasma concentrations of 5-MeO-DMT (**15**), without altering the formation of the active metabolite, bufotenin [133]. Finally, deuterium substitution of 5-MeO-DMT (**15**), which results in reduced metabolism, increased its behavioral effects in rats to a profile similar to that evoked by ordinary 5-MeO-DMT (**15**) in conjunction with MAO-A inhibition [56].

We have attempted to investigate the basis of Ayahuasca polypharmacology in PET studies using the hallucinogen 5-[^11^C]MeO-DMT (**47**), instead of labelled DMT (**9**). There was rapid uptake of 5-[^11^C]MeO-DMT (**47**) in brain of anesthetized pigs, but with little evidence of heterogenous distribution matching that expected for a 5HT_2A_ ligand; indeed, efforts to increase its uptake in pig brain by pretreatment with the MAO inhibitor pargyline (Figure 7) or displace its binding in pig brain with ketanserin (**7**) were unsuccessful. We were unable to account for the negative results, but suppose that the B_max_/K_D_ ratio may simply have been too low to impart a measurable BP_ND_. This pharmacological identity of the “hotspot” in pig ventral striatum remains unknown.

## 7. Effects of Hallucinogens on Energy Metabolism and Cerebral Blood Flow

### 7.1. Cerebral Glucose Metabolic Rate

Agonism at serotonin receptors might have intrinsic effects on neuronal energy metabolism. The 2-[^14^C]deoxyglucose autoradiographic method gives qualitative or quantitative estimates of the local rate of cerebral glucose consumption (CMRglc) based on the semi-irreversible metabolic trapping of the tracer in living cells. Administration to awake (but immobilized) rats of LSD (**1**) (15 or 150 µg/kg, i.v.) evoked widespread 10–30% reductions in CMRglc, notably throughout neocortex, and in thalamus, lateral geniculate, the basal ganglia, and the raphé nuclei, but with little effect in cerebellum [134]. In that study, treatment with 5-MeO-DMT (**15**, 0.2 or 2 mg/kg, i.v.) provoked similar patterns of reduced CMRglc, although generally of lower magnitude than the reductions seen with LSD (**1**). These rather widespread reductions of CMRglc stand in contrast to findings with selective agonists of 5HT_1A_ receptors (1 mg/kg 8-hydroxy-2-(di-*N*-propylamino)tetralin) or 5HT_1B_ receptors (3 mg/kg 5-methoxy 3-(1,2,3,6-tetrahydro-4-pyridinyl)-1*H* indole succinate). Both of those treatments evoked *increased* 2-[^14^C]-deoxyglucose trapping in cerebellum and motor cortex, and decreases in hippocampus, whereas the 5HT_2B_ agonist also evoked large increases in the basal ganglia [135]. Thus, one might suppose that net effects of hallucinogens on cerebral glucose metabolism results from drug actions at multiple serotonin receptor types, which may have opposing individual effects. Clearly, there is a need to undertake quantitative CMRglc studies with other hallucinogens of greater subtype specificity than is afforded by LSD (**1**) or 5-MeO-DMT (**15**). In addition, gender differences in response to psilocin [136] or the stress due to immobilization may also have been factors influencing the effects on CMRglc in rat brain.

The effect of psilocybin (**2**) (15 or 20 mg, p.o.) on CMRglc was tested in a human [^18^F]-fluorodeoxyglucose (FDG) PET study, in which post-drug findings were compared with the volunteers’ own baseline PET recordings [137]. Results in a mixed gender group of (*n* = 10) healthy volunteers indicated a global increase in CMRglc after treatment, which was most distinct (+25%) in the frontal cortex, anterior cingulate and medial temporal cortex, with lesser increases (+15%) in the basal ganglia and sensorimotor and occipital cortical regions. The increases in CMRglc were bilateral and roughly symmetrical, but with a tendency for more pronounced increases in the right hemisphere. Several of the regional increases correlated positively with scores in “hallucinatory ego disintegration”, which lead the authors to compare psilocybin (**2**) effects with the hypermetabolism in frontal cortex reported during acute exacerbations in patients with chronic schizophrenia.

Another human PET study investigated FDG uptake in eight subjects starting 90 min after administration of psilocybin (**2**) (0.2 mg/kg, p.o.), with quantitation relative to an arterialized venous blood input function. Relative to a control group, the treatment provoked 5–10% increases in CMRglc in frontal operculum, anterior cingulate cortex, and inferior temporal cortex, along with 5–10% decreases in the precentral cortex and thalamus in a study by Gouzoulis-Mayfrank et al. [138]. Whereas Vollenweider et al. [137] reported symmetrical changes in CMRglc, Gouzoulis-Mayfrank et al. [138] saw greater effects in the right hemisphere. They also reported significant correlations between scores in psychometric tests with CMRglc changes in several brain regions. There was relatively little overlap between the patterns of altered CMRglc in subjects receiving psilocybin (**2**) versus the non-hallucinogenic indirect serotonin/dopamine agonist methylendioxymetamphetamine (MDMA). This may relate to the greater pharmacological specificity of psilocybin (**2**), which is a mixed 5HT_2_/5HT_1_ agonist. However, that study calculated psilocybin-induced CMRglc changes relative to values in an untreated control group [138]; comparison relative to own baseline values as in the earlier study [137] might have given a more sensitive indication of the pattern of psilocybin effects.

### 7.2. Cerebral Perfusion

PET studies with [^15^O]-water in anesthetized pig did not indicate any effects of LSD (**1**) treatment on cerebral perfusion [98]. In contrast, arterial spin labeling (ASL) measurements of cerebral perfusion indicated that LSD (**1**) increased CBF in visual cortex of human volunteers [139]. The increased CBF and the extent of the functional connectivity of the visual cortex correlated with the intensity of visual hallucinations; we suppose that anesthesia may have interfered in the effects of LSD (**1**) on CBF in the pig PET study. Furthermore, we did not measure plasma levels of LSD (**1**) in the pig, so we cannot exclude that possibility of mis-injection of LSD (**1**), perhaps due to adherence the drug to plastic syringes.

Perfusion SPECT studies in a group of (*n* = 12) healthy men showed that administration of mescaline evoked reduced blood flow in the frontal lobe, especially on the right side, which was correlated with impairment in a face/non-face task associated with the right frontal lobe [140]. A group of 15 healthy males with previous hallucinogen experience had perfusion SPECT scans before and after receiving authentic Ayahuasca at a dose of 1 mg DMT (**9**)/kg [141]. In contrast to results for mescaline, the Ayahuasca treatment provoked bilateral activations of perfusion in the anterior insula (which were more pronounced in the right hemisphere), along with activation in the right anterior cingulate/frontomedial cortex, and the left amygdala. These activations coincided with distinct increases in score in a hallucinogen rating scale. The authors interpreted these findings in functional anatomic terms as indicative of increased interoception and somatic awareness (insula), motivational state (cingulate) and emotional arousal (amygdala). This increased right side perfusion after Ayahuasca [141] and the right side hypermetabolism after psilocybin [137] seem at odds with the right side hypoperfusion after mescaline [140]. These discrepancies could well relate to differing actions of the hallucinogens, which may be distinct with respect to effects on flow-metabolism coupling. This could result in the generally opposite effects on perfusion and glucose consumption.

In a clinical trial of Ayahuasca, a group of 17 patients with recurrent major depressive disorder received a single dose of authentic Ayahuasca containing approximately 120 mg DMT (**9**) and 32 mg harmine (**44**) [142]. The investigators administered various psychometric rating scales in the hours and weeks after the single treatment, and recorded SPECT scans for CBF at baseline and at eight hours after the drug administration. The score in the Clinician Administered Dissociative States Scale (CADSS) peaked at about 80 min after drug administration, and the mean Hamilton depression rating scales declined from 25 to 15 in the three hours after administration, and remained around 10 during the following three weeks. Parametric mapping of CBF changes indicated increases in some very small clusters (<50 voxels) localized to the left nucleus accumbens, right insula, and left cingulate cortex, which the authors felt to be regions activated by antidepressant treatments. While similar in distribution, the magnitude of the CBF response in depression patients was distinctly less than that in the study of healthy volunteers by Riba et al. [141]. This could indicate a functional inactivation of serotonin receptors in the depressed state.

In a case study using SPECT, a patient suffering from alcohol dependence underwent a trial treatment with ibogaine hydrochloride (**17** HCl, 1.5 g) followed two days later with vaporized 5-MeO-DMT (**15**, 5–7 mg) [143]. The patient experience insightful visions during the ibogaine (**17**) treatment, and later reported a decline in alcohol consumption. Relative to baseline perfusion SPECT imaging, there were post-treatment increases in brain perfusion in bilateral caudate nuclei, left putamen, right insula, as well as temporal and occipital cortex and cerebellum.

In recent years, functional MRI (fMRI) measures of perfusion with arterial spin labelling (ASL) have come to supersede SPECT and PET methods. In one such study, mixed gender groups of (*n* = 29) volunteers underwent an ASL examination of cerebral perfusion after treatment with psilocybin at a low dose (0.16 mg/kg) or high dose (0.22 mg/kg) [144]. The psilocybin treatments evoked absolute reductions in perfusion in widespread brain regions including the neocortex, insula, hippocampus, and striatum; the authors showed that global normalization resulted in apparent increases in perfusion in frontal regions. This seems generally consistent with findings in another fMRI study, which showed decreased BOLD signal and absolute perfusion in hub regions including the thalamus and cingulate cortex of (*n* = 15) healthy volunteers treated with psilocybin [145]. Interestingly, these same brain regions have a notably dense serotonin innervation; hybrid scanning with fMRI-ASL in conjunction with serotonin receptor PET imaging may held to establish the relationship between hallucinogen binding sites and cerebral perfusion. The ASL findings generally concur with SPECT perfusion studies in healthy controls showing reduced cerebral perfusion after treatment with hallucinogens. Since this review focusses on molecular brain imaging, we refer readers elsewhere for a comprehensive review of fMRI studies on hallucinogen actions [8].

## 8. General Conclusions

There is general agreement that the hallucinogenic compounds reviewed herein share the property of agonism at 5HT_2A_ sites, but some compounds also have high affinity at other serotonin receptor subtypes, notably 5HT_2C_ and 5HT_1A_. Indeed, 5-MeO-DMT (**15**) has considerably higher affinity at 5HT_1_-like sites compared to 5HT_2_-like sites in vitro [56], and 5HT_1A_ and 5HT_2A/B_ antagonists are equally effective in blocking its behavioral effects in rats [133]. While the 5HT_2A_-prefering antagonist ketanserin (**7**) can relieve visual hallucinations, aspects of the experience evoked by some compounds may well be due to effects at other serotonin receptor subtypes, or in some cases through binding to dopamine receptors and plasma membrane transporters. Comparisons of affinity and selectivity of various compounds is sometimes difficult due to disagreement between results of displacement studies (Ki) and estimates of affinity (K_D_) in vitro. The possibility that hallucinogens may activate multiple second messenger systems pathways including adenylyl cyclase and phospholipase C adds an additional layer of complexity, especially considering that slight structural modifications of certain hallucinogens can attenuate hallucinogenic potency without necessarily modifying affinity at key receptor targets.

Early studies using radiolabeled hallucinogens such as [^14^C]-psilocin (**8**) confirmed that hallucinogenic compounds rapidly enter the brain, being relatively unhindered at the blood–brain barrier. This is not always the case; the low octanol:water partition coefficient for bufotenine (**10**) and the predicted low permeability to the blood–brain barrier, has been invoked to explain its relatively low hallucinogenic potency, despite moderate activity in serotonin receptor functional assays [146]. Most hallucinogens undergo two-phase metabolism, whereby de-alkylation precedes glucuronate conjugation; the pharmacokinetics of particular compounds can determine their pharmacodynamic responses. Tryptamine derivatives such as DMT (**9**) tend to undergo rapid metabolism, such that the hallucinogenic experience lasts only some minutes after administration, whereas LSD (**1**) has plasma half-life of several hours in humans. There is general agreement between the time course of hallucinogenic experiences and plasma concentrations of the relevant compound. Given this, we can expect that pharmacogenetic studies should reveal factors relating to individual vulnerability to hallucinogenic compounds. The Ayahuasca phenomenon presents an interesting case where treatment with non-hallucinogenic inhibitors of monoamine oxidase, but attenuating metabolism, augments the intensity and duration of the experience evoked by DMT (**9**).

Despite early success with *N*1-([^11^C]-methyl)-2-bromo-LSD ([^11^C]-MBL, **33**), there have been relatively few molecular imaging studies of radiolabeled hallucinogen analogues, either in humans or experimental animals. More commonly, hallucinogens serve as a pharmacological challenge to determine indirect effects on availability of dopamine D_2_ receptors labelled with [^11^C]-raclopride (**41**), or occupancy at serotonin receptors labelled with some other radiotracer. A very few PET studies have examined the effects of treatment with a hallucinogen on cerebral metabolism to FDG-PET or cerebral blood flow; the limited available data indicate hypermetabolism despite hypoperfusion, which implies that hallucinogens may alter flow-metabolism coupling in brain. Thus, there is an enormous scope to undertake PET and SPECT studies of hallucinogen binding at various receptor sites, and to examine the effects of acute and chronic exposure on cerebral metabolism and synaptic density markers.

## Figures and Tables

**Figure 1 molecules-26-02451-f001:**
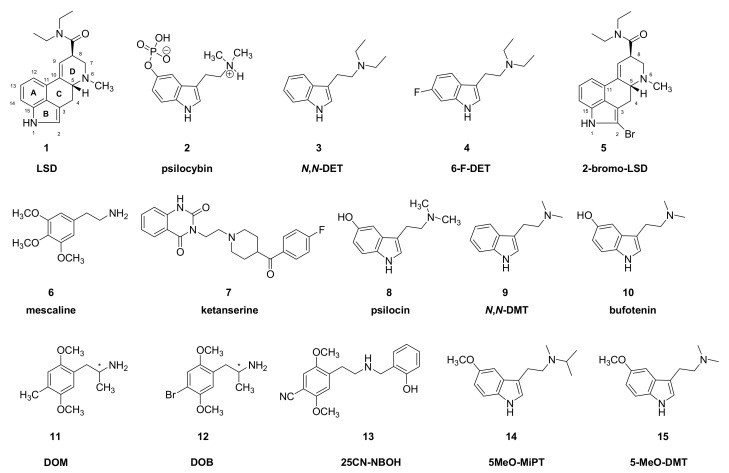
Chemical structures of LSD and some classic hallucinogenic compounds—(*) indicates chiral centers.

**Figure 2 molecules-26-02451-f002:**
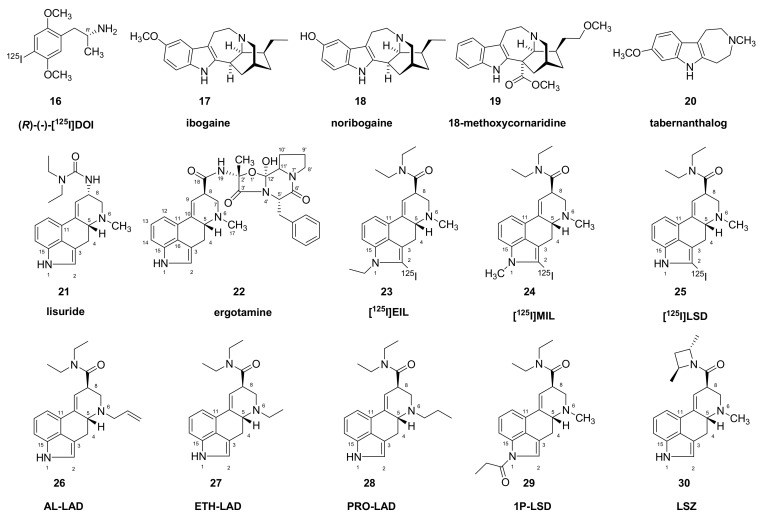
Structure of selected ibogaine and lisuride derivatives.

**Figure 3 molecules-26-02451-f003:**
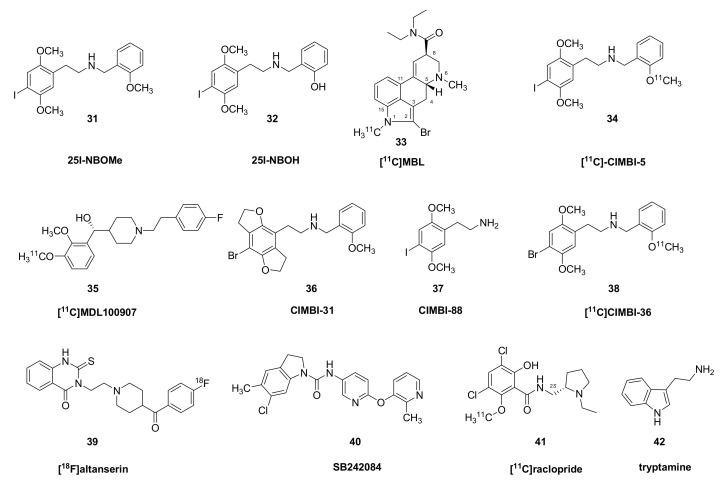
Structures of some dimethoxyamphetamines and serotonin receptor antagonists.

**Figure 4 molecules-26-02451-f004:**
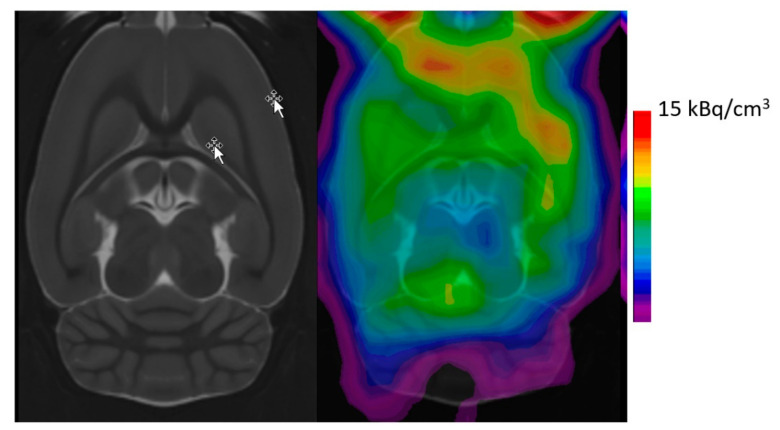
[^11^C]-Cimbi-36 (**38**) activity summed over the 45 min scan time in rat brain (**right**) overlaid on an MR template image in the horizontal plane (**left**). High activity in cortical regions, especially mPFC, moderate activity in striatal and hippocampal regions and low binding in cerebellum.

**Figure 5 molecules-26-02451-f005:**
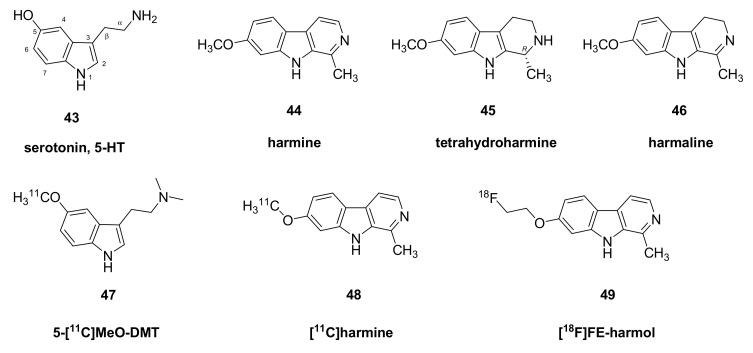
Structures of serotonin and some indole alkaloids.

**Figure 6 molecules-26-02451-f006:**
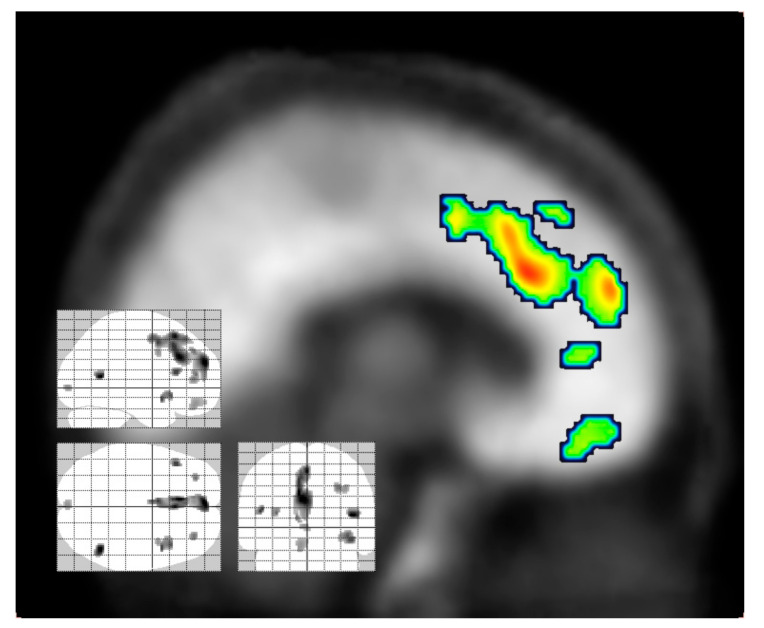
Inverse Correlation of 5D-ASC Global Scale scores and [^18^F]-altanserin (**39**) apparent distribution volume [DV’]. Results of a voxel based correlation analysis (Δ 5D-ASC global vs. ΔDV’, threshold *p* < 0.005, uncorrected) using Statistical Parametric Mapping [105]. Reproduced with permission.

**Figure 7 molecules-26-02451-f007:**
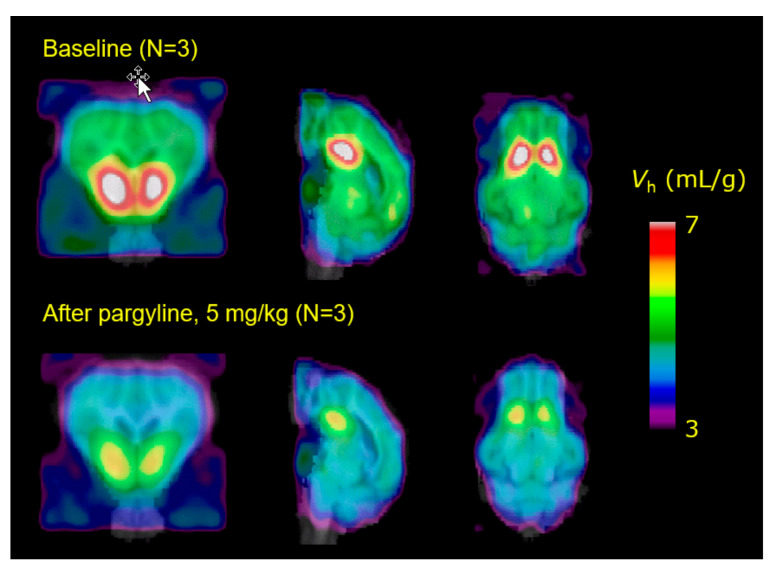
Mean parametric maps of the total distribution volume of 5-[^11^C]-MeO-DMT (**47**) in brain of groups of three pigs at baseline and after treatment with the irreversible MAO A/B inhibitor, pargyline. There was no evidence of potentiation of the tracer uptake by blockade of MAO, nor was the binding in ventral striatum displaceable in studies with ketanserin (**7**) pretreatment (Jensen and Cumming, unpublished observations).

## Data Availability

Not applicable.

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
