# Peer review of "Molecular and Functional Imaging Studies of Psychedelic Drug Action in Animals and Humans"

_molecules, 2021, doi:10.3390/molecules26092451_

Round 1

Reviewer 1 Report

Authors reviewed the psychological effects of many hallucinogens and presented their effects. This report is rare in discussing the potentials of hallucinogens and possible mechanisms by PET images from rat and pig. It will be more convincing if authors present the various possible mechanisms, metabolisms and other side effects in a series of flow diagrams and tables for the comparisons. There are too many run-on sentences and mix-ups in using verb tenses.

Author Response

We were unable to find many instances of run-on sentences. We are now more consistent in citing literature findings in the past tense.  We are unsure what the reviewer finds unconvincing about our review, which seems to be the only comprehensive review of molecular imaging studies of hallucinogen action. In general, the available results of binding studies, ex vivo uptake studies and PET/SPECT findings for the various hallucinogens are so fragmentary that we prefer to avoid presenting results in tables.

Reviewer 2 Report

This review gives an overview of the pharmacology of psychedelic compounds in order to provide a foundation for understanding human and animal PET and SPECT imaging data related to these drugs.  As psychedelics and related compounds are increasingly being considered for use in the clinic, it will be critical to use modern imaging methods to accurately assess target engagement.  Thus, this review is very timely.  Overall, I found the review to be accurate and straightforward.  I recommend publication with a few minor suggestions:

  • Figure 1. I’m assuming that the star indicates that these compounds are racemic mixtures, but you might want to explicitly state this in the figure legend.

  • Line 79. The structural classes of serotonergic hallucinogens are more appropriately described as ergolines, tryptamines, and phenethylamines (or amphetamines if you want to exclude mescaline).

  • Line 121. The prevalence of HPPD is highly debated, and there is no indication that plasticity mechanisms lead to HPPD. Thus, please remove the comment speculating that plasticity might lead to HPPD.

  • Line 188. There is no 5-HT1C receptor. Do the authors mean 5-HT2C?

  • Line 252. “reflect” should be “reflex”

  • Line 262. Psilocybin is the phosphoryloxy compound. Psilocin is the free phenol.  I think the authors got these two backwards in this sentence.

  • Line. 264. The authors claim that psilocin has no affinity for 2B receptors, but that is not true. The PDSP has a report of psilocin having affinity for 2B ~450 nM.  Psilocin might be more selective for 2A and 2C, but it still hits 2B.  Given the importance of this receptor in cardiac valvulopathy, this point must be made.

  • Line 313. The authors should mention the work of Gonzalez-Maeso indicating that a putative 5-HT2A-mGlu2 heterodimer might be responsible for the hallucinogenic effects of psychedelics given that lisuride does not activate the dimer.

  • Identification of a serotonin/glutamate receptor complex implicated in psychosis

  • line 337. Again, do the authors mean 5-HT2C? There is no 5-HT1C as it was reclassified.

  • Imaging studies for a wide variety of 2-iodo ergolines were discussed, but very little was mentioned about their pharmacological properties. Are these agonists or antagonists at 5-HT2A receptors?

  • Line 583. Additional papers demonstrating in vivo increases in spine density following psychedelic administration include:

“Psychedelics Promote Structural and Functional Plasticity”

“Psilocybin induces rapid and persistent growth of dendritic spines in frontal cortex in vivo”

  • It is puzzling that [14C]deoxyglucose studies (LSD and 5-MeO-DMT) produce opposite results as 18F-fluorodeoxyglucose studies (psilocybin). The authors should comment on the discrepancy in more detail.

  • The discussion of cerebral blood flow is weak. This is especially important given the known ability of psychedelics to induce vasoconstriction and the fact that psilocybin and LSD have shown promise for treating migraines and cluster headaches.

Author Response

Figure 1. I’m assuming that the star indicates that these compounds are racemic mixtures, but you might want to explicitly state this in the figure legend.  (*) indicates chiral centres, as now stated in the figure caption

Line 79. The structural classes of serotonergic hallucinogens are more appropriately described as ergolines, tryptamines, and phenethylamines (or amphetamines if you want to exclude mescaline). Corrected

Line 121. The prevalence of HPPD is highly debated, and there is no indication that plasticity mechanisms lead to HPPD. Thus, please remove the comment speculating that plasticity might lead to HPPD.

The text is amended to read “Finally, the phenomena of “flashbacks” and hallucinogen-persisting perception disorder (HPPD) sometimes occurring in LSD users may call for reassessing the notion that visual hallucinations are only due to acute activation of serotonin receptors [1]”. We have removed the speculation about neuroplasticity (although any persistent perceptual change almost certainly requires the evocation of neuroplasticity).

Line 188. There is no 5-HT1C receptor. Do the authors mean 5-HT2C? The obsolete nomenclature is corrected throughout.

Line 252. “reflect” should be “reflex” Corrected

Line 262. Psilocybin is the phosphoryloxy compound. Psilocin is the free phenol.  I think the authors got these two backwards in this sentence. Indeed, we had switched the compounds in this sentence. Now corrected.

Line. 264. The authors claim that psilocin has no affinity for 2B receptors, but that is not true. The PDSP has a report of psilocin having affinity for 2B ~450 nM.  Psilocin might be more selective for 2A and 2C, but it still hits 2B.  Given the importance of this receptor in cardiac valvulopathy, this point must be made.

We have added the phrase, “Screening indicated a moderate affinity for psilocin at 5HT2B sites [2], which may have some bearing on the propensity to cause cardiac valvulopathy”.

Line 313. The authors should mention the work of Gonzalez-Maeso indicating that a putative 5-HT2A-mGlu2 heterodimer might be responsible for the hallucinogenic effects of psychedelics given that lisuride does not activate the dimer.

We have added the text, “Furthermore, the 5HT2A receptor forms a functional heterodimer with the mGluR2 receptor, which evokes allosteric effects on serotonin agonist binding [3]; this interaction reduces the hallucinogen-specific Gi/o protein signalling and behaviour and may account for the lack of hallucinogenic action of psilocin noted above. Certainly, the 5HT2A/mGluR2 dimer adds an additional level of complexity to the mechanism of action of hallucinogens.”

line 337. Again, do the authors mean 5-HT2C? There is no 5-HT1C as it was reclassified. Corrected globally.

Imaging studies for a wide variety of 2-iodo ergolines were discussed, but very little was mentioned about their pharmacological properties. Are these agonists or antagonists at 5-HT2A receptors?

We have added the text and citations: “Early research into structure-function indicated that methylation of ergotamines at the N1 position enhances serotonin antagonism in the isolated rat uterus assay [4], while decreasing hallucinogenic potency predicted from a quantitative structure-activity relationship (QSAR) study [5]. 

Line 583. Additional papers demonstrating in vivo increases in spine density following psychedelic administration should be included.

We have added the following text: “In a recent rodent study, a single dose of DOI exerted rapid effects on the dendritic spine structure in frontal cortex that were dependent on 5HT2A receptors [6]. The same treatment evoked chromatin changes that persisted long after the DOI treatment, suggesting that long term effects of therapeutic hallucinogen treatment may arise through persistent and epigenetic changes in synaptic architecture. Others have recently used chronic two-photon microscopy to monitor synaptic spine density in mouse cortex [7]. In female mice, a single dose of psilocybin (1 mg/kg, i.p.) evoked a 20% increase in layer V spine density that persisted for up to 34 days, along with a sharp increase in spine dimension that peaked at day one and declined over the following week; these changes were scarcely evident in male mice.  Overall, these structural studies concur with the SV2A results discussed above and fit into a general framework whereby rapid antidepressant responses to ketamine also occur along with dendritic expansion, albeit by a different mechanism [8]. “

It is puzzling that [14C]deoxyglucose studies (LSD and 5-MeO-DMT) produce opposite results as 18F-fluorodeoxyglucose studies (psilocybin). The authors should comment on the discrepancy in more detail.

We agree that this is puzzling, and note that it is unfortunate that there have been so few studies of the effects of hallucinogens on glucose metabolism. We added the following text: “Thus, one might suppose that net effects of hallucinogens on cerebral glucose metabolism results from drug actions at multiple serotonin receptor types, which may have opposing effects. Clearly, there is a need to undertake quantitative CMRglc studies with other hallucinogens of greater subtype specificity than is afforded by LSD or 5-MeO-DMT. In addition, gender differences in response to psilocin [9] the stress due to immobilization may also have been a factor influencing the effects on CMRglc in rat brain.”   

The discussion of cerebral blood flow is weak. This is especially important given the known ability of psychedelics to induce vasoconstriction and the fact that psilocybin and LSD have shown promise for treating migraines and cluster headaches.

We respectfully disagree that the efficacy of psilocybin and LSD for alleviating migraine and cluster headache is relevant to the present topic of cerebral perfusion. Migraine attacks are associated with phasic changes (increases and decreases) in cerebral perfusion [10], which might be alleviated by hallucinogens. However, we favor a model in which trigeminal innervation of the MCA [11] mediates pain and cerebral perfusion changes occurring during migraine. This pathway of vascular control seems distinct from the neurovascular coupling in the brain parenchyma. 

We have expanded the review of perfusion studies by adding the following text on page 22:

Perfusion SPECT studies in a group of (n=12) healthy men showed that administration of mescaline evoked reduced blood flow in the frontal lobe, especially on the right side, which was correlated with impairment in a face/non-face task associated with the right frontal lobe  [12].”

“This increased right side perfusion after Ayahuasca [13] and the right side hypermetabolism after psilocybin [14] seem at odds with the right side perfusion hypoperfusion after mescaline [12]. While these discrepancies could well relate to differing actions of the hallucinogens, it is also possible that altered flow-metabolism coupling could result in opposite effects on perfusion and glucose consumption.

We add an additional paragraph as follows:

“In recent years, functional MR measures of perfusion with arterial spin labelling (ASL) have come to supersede SPECT and PET methods. In one such study, mixed gender groups of (n=29) volunteers underwent an ASL examination of cerebral perfusion after treatment with psilocybin at a low dose (0.16 mg/kg) or high dose (0.22 mg/kg) [15]. The psilocybin treatments evoked absolute reductions in perfusion in widespread brain regions including the neocortex, insula, hippocampus, and striatum; the authors showed that global normalization resulted in apparent increases in perfusion in frontal regions.  This seems generally consistent with findings in another fMRI study, which showed decreased perfusion and BOLD signal in hub regions including the thalamus and cingulate cortex of (n=15) healthy volunteers treated with psilocybin [16]. Interestingly, these same brain regions have a notably dense serotonin innervation; hybrid scanning with fMRI-ASL in conjunction with serotonin receptor PET imaging may held to establish the relationship between hallucinogen binding sites and cerebral perfusion. Since this review focusses on molecular brain imaging, we refer readers elsewhere for a comprehensive review of fMRI studies on hallucinogen actions [17].”

REFERENCES

  1. Orsolini, L., et al., The "Endless Trip" among the NPS Users: Psychopathology and Psychopharmacology in the Hallucinogen-Persisting Perception Disorder. A Systematic Review. Front Psychiatry, 2017. 8: p. 240.
  2. Lee, H.M. and B.L. Roth, Hallucinogen actions on human brain revealed. Proc Natl Acad Sci U S A, 2012. 109(6): p. 1820-1.
  3. Gonzalez-Maeso, J., et al., Identification of a serotonin/glutamate receptor complex implicated in psychosis. Nature, 2008. 452(7183): p. 93-7.
  4. Cerletti, A. and W. Doepfner, Comparative study on the serotonin antagonism of amide derivatives of lysergic acid and of ergot alkaloids. J Pharmacol Exp Ther, 1958. 122(1): p. 124-36.
  5. Gupta, S.P., Singh, P., Bindal, M.C., QSAR studies on hallucinogens. Chem. Rev., 1983. 83: p. 633-649.
  6. de la Fuente Revenga, M., Zhu, B., Guevara, C.A., Naler, L.B., Saunders, J.B., Zhou, Z., Toneatti, R., Sierra, S., Wolstenholme J.T., Beardsley, P.M., Huntley, G.W., Lu, C., Gonzalez-Maeso, J. , Prolonged epigenetic and synaptic plasticity alterations following single exposre to a psychedelic in mice. 2021.
  7. Shao, L.-X., Liao, C., Gregg, I., Savalia, N.K., Delagarza, K., and Kwan, A.C., Psilocybin iduces rapid and persistent growth of dendritic spines in frontal cortex in vivo. 2021.
  8. Savalia, N.K., L.X. Shao, and A.C. Kwan, A Dendrite-Focused Framework for Understanding the Actions of Ketamine and Psychedelics. Trends Neurosci, 2021. 44(4): p. 260-275.
  9. Tyls, F., et al., Sex differences and serotonergic mechanisms in the behavioural effects of psilocin. Behav Pharmacol, 2016. 27(4): p. 309-20.
  10. Bartolini, M., et al., Cerebral blood flow changes in the different phases of migraine. Funct Neurol, 2005. 20(4): p. 209-11.
  11. Warfvinge, K., et al., Estrogen receptors alpha, beta and GPER in the CNS and trigeminal system - molecular and functional aspects. J Headache Pain, 2020. 21(1): p. 131.
  12. Hermle, L., E. Gouzoulis-Mayfrank, and M. Spitzer, Blood flow and cerebral laterality in the mescaline model of psychosis. Pharmacopsychiatry, 1998. 31 Suppl 2: p. 85-91.
  13. Riba, J., et al., Increased frontal and paralimbic activation following ayahuasca, the pan-Amazonian inebriant. Psychopharmacology (Berl), 2006. 186(1): p. 93-8.
  14. Vollenweider, F.X., et al., Positron emission tomography and fluorodeoxyglucose studies of metabolic hyperfrontality and psychopathology in the psilocybin model of psychosis. Neuropsychopharmacology, 1997. 16(5): p. 357-72.
  15. Lewis, C.R., et al., Two dose investigation of the 5-HT-agonist psilocybin on relative and global cerebral blood flow. Neuroimage, 2017. 159: p. 70-78.
  16. Carhart-Harris, R.L., et al., Neural correlates of the psychedelic state as determined by fMRI studies with psilocybin. Proc Natl Acad Sci U S A, 2012. 109(6): p. 2138-43.
  17. Dos Santos, R.G., et al., Classical hallucinogens and neuroimaging: A systematic review of human studies: Hallucinogens and neuroimaging. Neurosci Biobehav Rev, 2016. 71: p. 715-728.